# Re-evaluating randomized clinical trials of psychological interventions: Impact of response shift on the interpretation of trial results

**M. G. E. Verdam**[1,2]*, **W. van Ballegooijen**[3], **C. J. M. Holtmaat**[3], **H. Knoop**[1,4], **J. Lancee**[5], **F. J. Oort**[6], **H. Riper**[3,7], **A. van Straten**[3,7], **I. M. Verdonck-de Leeuw**[3,7,8], **M. de Wit**[7], **T. van der Zweerde**[3], **M. A. G. Sprangers**[1]

1 Department of Medical Psychology, Amsterdam University Medical Centers, Amsterdam Public Health Research Institute, University of Amsterdam, Amsterdam, The Netherlands, 2 Department of Methodology and Statistics, Institute of Psychology, Leiden University, Leiden, The Netherlands, 3 Department of Clinical, Neuro-, & Developmental Psychology, Faculty of Behavioural and Movement Sciences, Vrije Universiteit Amsterdam, Amsterdam, The Netherlands, 4 Department of Medical Psychology, Expert Center for Chronic Fatigue, Amsterdam Public Health Research Institute, VU University Medical Center, Amsterdam, The Netherlands, 5 Department of Clinical Psychology, Faculty of Social and Behavioural Sciences, University of Amsterdam, Amsterdam, The Netherlands, 6 Research Institute of Child Development and Education, University of Amsterdam, Amsterdam, The Netherlands, 7 Mental Health, Amsterdam Public Health Research Institute, Amsterdam, The Netherlands, 8 Department of Otolaryngology-Head and Neck Surgery, Amsterdam University Medical Centers, Vrije Universiteit Amsterdam, Amsterdam, The Netherlands

* m.g.e.verdam@amsterdamumc.nl

## Abstract

### Background

Effectiveness of psychological treatment is often assessed using patient-reported health evaluations. However, comparison of such scores over time can be hampered due to a change in the meaning of self-evaluations, called 'response shift'. Insight into the occurrence of response shift seems especially relevant in the context of psychological interventions, as they often purposefully intend to change patients' frames of reference.

### Aims

The overall aim is to gain insight into the general relevance of response shift for psychological health intervention research. Specifically, the aim is to re-analyse data of published randomized controlled trials (RCTs) investigating the effectiveness of psychological interventions targeting different health aspects, to assess (1) the occurrence of response shift, (2) the impact of response shift on interpretation of treatment effectiveness, and (3) the predictive role of clinical and background variables for detected response shift.

### Method

We re-analysed data from RCTs on guided internet delivered cognitive behavioural treatment (CBT) for insomnia in the general population with and without elevated depressive symptoms, an RCT on meaning-centred group psychotherapy targeting personal meaning

**Data Availability Statement:** The data that were used for the analyses presented in the current

paper are available through the first author's Open Science Framework page at https://osf.io/fv8c5/.

**Funding:** This research was supported by a grant from the Mental Health Alliance Fund provided by the Amsterdam Public Health Research Institute (APH; http://www.amsterdam-public-health.org). Grant registration number Amsterdam UMC: V.2018.29. The funder had no role in study design, data collection and analysis, decision to publish, or preparation of the manuscript.

**Competing interests:** The authors have declared that no competing interests exist.

for cancer survivors, and an RCT on internet-based CBT treatment for persons with diabetes with elevated depressive symptoms. Structural equation modelling was used to test the three objectives.

## Results

We found indications of response shift in the intervention groups of all analysed datasets. However, results were mixed, as response shift was also indicated in some of the control groups, albeit to a lesser extent or in opposite direction. Overall, the detected response shifts only marginally impacted trial results. Relations with selected clinical and background variables helped the interpretation of detected effects and their possible mechanisms.

## Conclusion

This study showed that response shift effects can occur as a result of psychological health interventions. Response shift did not influence the overall interpretation of trial results, but provide insight into differential treatment effectiveness for specific symptoms and/or domains that can be clinically meaningful.

## Introduction

Psychological interventions play an important role in the promotion of good health, which The World Health Organization (WHO) defines as a 'state of complete physical, mental and social well-being and not merely the absence of disease or infirmity'. That is, psychological interventions aim to induce behavioral, cognitive and/or emotional changes that are targeted to help attain good health and therefore improve general wellbeing and quality of life [cf. 1, 2].

Effectiveness of psychological interventions is often assessed through self-assessment. That is, individuals are asked to rate their own level of functioning or health. Comparison of such self-assessed health scores over time between treatment- and control groups are used to examine treatment effectiveness. However, the evaluation and interpretation of treatment effectiveness can be hampered due to the occurrence of 'response shift' [3]. Response shift refers to a change in the meaning of self-evaluations, that can potentially invalidate the comparison of self-evaluations over time. Sprangers and Schwartz [4] distinguish three types of response shift. *Recalibration* refers to a change in respondents' internal criteria with which they assess the construct of interest. For example, after intervention, the response "occasionally" on an item about feelings of sadness may represent another level of depression than before intervention because the meaning of "occasionally" has changed. *Reprioritization* refers to a change in the relative importance of subdomains or aspects of the construct of interest. For example, after intervention, the scores on an item about perceiving to be disliked by others may become less important to the measurement of depression. *Reconceptualization* refers to a change in the meaning of the target construct. For example, after intervention, the concept of "depression" may include somatic components (e.g. sleep problems) that were not part of the meaning of the concept of depression for the same respondents before intervention. When response shift occurs, changes over time are not just indicative of changes in health, but also of changes in the meaning of the self-evaluations. That is, people with similar health over time may nevertheless score differently on health-measures because the meaning of their self-evaluations has changed. Or, vice versa, similar scores on health-measures are not necessarily indicative of

stable health, but instead are the result of a change in meaning. In other words, response shifts may contaminate treatment effects (reduce or amplify), and therefore need to be taken into account when evaluating and interpreting treatment effectiveness.

Psychological interventions aimed at improving health may induce response shifts as they often purposefully intend to change patients' frames of reference and values. For example, psychoeducation may be intended to change patients' interpretations of (the severity of) events, and cognitive restructuring in CBT is specifically aimed to change patients' thinking patterns. Hence, response shifts can also be considered beneficial treatment effects [5, 6]. Insight into the occurrence of response shifts is thus not only important for the valid and detailed evaluation of intervention effects but may also enhance our understanding of the effectiveness of interventions.

Literature on the investigation of response shift focusses mostly on the effect of major health events and/or medical interventions, e.g. the occurrence of response shift after surgery or radiotherapy in patients with cancer [see 7–9 for overviews]. To our knowledge, these are the only studies that investigated the occurrence of response shift as a result of psychological interventions. The first investigated the occurrence of response shift in measures of depression from a randomized clinical trial (RCT) that included both medical and psychotherapy treatment groups [10]. They found substantial response shift, particularly in the psychotherapy groups, and concluded that these results may confound measures of treatment efficacy. Another study [11] showed that the occurrence of recalibration response shift inflated treatment effectiveness in measures of depression from students who received CBT from their school counsellors. Smith et al. [12] detected several indications of response shift in self-report measures of psychological distress from help-seeking problem gamblers who received CBT, but did not report on the impact of response shift on estimated treatment effect. Finally, Carlier at al. [13] investigated response shift in self-reported psychological distress from psychiatric outpatients who received psychotherapy (mainly CBT), pharmacotherapy, or a combination of both. They found several indications of reprioritization and reconceptualization, but these response shifts did not impact the assessment of treatment effectiveness. In summary, in all four studies response shift was detected, but the results regarding the impact of response shift on treatment effectiveness were mixed. Also, due to heterogeneity in the type of interventions included in the study by Carlier et al. [13], and a lack of control conditions in all four studies, it is difficult to determine whether response shift is induced by psychological interventions or occurs as a result of the natural course of time. Moreover, these studies did not investigate variables that could predict the occurrence of response shift. Thus, although some evidence of the occurrence of response shift as a result of psychological interventions exist, there is still a lack of knowledge about treatment-induced response shift in the context of psychological interventions, whether or how they affect the interpretation of treatment effectiveness, and if it can be predicted which respondents are especially prone to experience response shift.

The overall objective of this study is to use secondary data analysis of published randomized controlled trials (RCTs) investigating the effectiveness of psychological interventions targeting different health aspects, to investigate the occurrence of response shift. This study was intended as a proof-of-principle, in order to gain insight into the general relevance of response shift for psychological health intervention research. Therefore, we have used available datasets that cover different patient groups and psychological interventions. The selected sample consisted of data from RCTs on guided internet delivered cognitive behavioural treatment (CBT) for insomnia in the general population with [14] and without [15] elevated depressive symptoms, an RCT on meaning-centered group psychotherapy targeting personal meaning for cancer survivors [16], and from an RCT on guided internet-based CBT treatment for persons with

diabetes with elevated depressive symptoms [17]. All these RCTs indicated that the interventions are effective with medium to large effect sizes using conventional analyses, e.g. (multilevel) regression analyses, that do not take into account response shift. For the current study, we formulate the following hypotheses. First, we hypothesize that response shifts occur in the primary self-reported outcomes. As we hypothesize that response shift is induced by psychological treatment, we expect response shift to occur only in the intervention groups, and not in the control groups. Second, the detected response shifts are hypothesized to be clinically meaningful in that they substantially impact the assessment of change. That is, when response shift is taken into account, we expect the treatment effects to be different (reduced or amplified). Moreover, as a third aim of this study, we will investigate the potential predictive role of clinical and background variables for the occurrence of response shift.

## Method

We re-analysed data from the four above mentioned published RCTs. This convenience sample data-sets were selected based on the following criteria: 1) the RCT results indicate effectiveness of the psychological intervention; and thus 2) response shift may have occurred; 3) total number of included patients > 100; and 4) availability of clinical and background information that can be included as explanatory variables. Because we are interested in mechanisms of change induced by psychological treatment, we excluded individuals in the treatment groups who did not follow sufficient intervention sessions (as defined in close collaboration with the respective trial-authors; see for more details below), and individuals in the control groups who indicated to have received (other) psychological treatment at post-assessment. Assessments in all trials were performed before randomization (baseline) and after the intervention or waiting period (post-assessment). We only included participants that completed both assessments. All trials were approved by the relevant Medical Ethical Committee, i.e. the Ethics Review Board of the University of Amsterdam (registration no. 2016-CP-7263), the Medical Ethics Committee of the VU Medical Centre Amsterdam (registration no. 2015/258), the Medical Ethics Committee of the Leiden University Medical Centre (registration no. P10.241), and the Medical Ethics Committee of the VU University Medical Centre (Registration no. 2007/47) respectively. Patients provided written informed consent prior to their participation (see the protocol papers [18–20] and Trial Register NTR6049 for more details regarding the methods of the four trials).

### CBT for insomnia dataset

**Patients.** The CBT for insomnia dataset consisted of two combined datasets. The first trial [14] consisted of 104 adult patients with insomnia disorder and suffering from (subclinical) symptoms of depression, randomized to a 5-session guided online treatment 'i-Sleep' including sleep diary monitoring (n = 52) or control group (n = 52; sleep diary monitoring control group without intervention). The second trial [15] consisted of a total of 134 adult patients with insomnia disorder recruited through general practices who were randomized to the same 5-session guided online i-Sleep treatment (n = 69) or control group (n = 65; care-as-usual control group without further interference). As both trials offered the same intervention, and were assumed to induce similar processes of change, we combined data from both trials to increase sample size; we refer to the combined dataset as the 'CBT for insomnia dataset'. Assessments were performed online, before randomization (baseline) and 8 [15] or 9 weeks [14] after the intervention or waiting period (post-assessment). For more information regarding in- and exclusion criteria and recruitment methods see [14, 15] respectively. The combined dataset consisted of 121 participants in the intervention group, of whom 91 participants (75%) fulfilled

the criteria of 'completer' (followed 4 out of 5 sessions; as this was considered to be an adequate dose of CBT as no new information is offered in session 5), and 77 of those 91 participants completed post-assessment. The control group consisted of 117 participants, of whom 2 participants (2%) were excluded because they reported to have received an alternative psychological treatment, and a total of 86 participants completed post-assessment.

**Intervention.** The online intervention i-Sleep consisted of five sessions of CBT for insomnia [20, 21]. The five sessions focused on (1) sleep hygiene and lifestyle, (2) stimulus control and sleep restriction therapy (in which the time in bed is restricted to the average time slept in the last week (with a 5-h minimum), with the aim of increasing the time in bed spent asleep and decreasing time in bed spent awake, (3) relaxation, (4) cognitive therapy addressing dysfunctional thoughts about sleep, and (5) relapse prevention [20]. Online guidance entailed feedback on exercises, discussing sleep restriction therapy/bedtimes based on the diary patients had to keep, and motivating participants to persevere the treatment. Patients in the control condition were offered i-Sleep after completing post-assessment [14] or 6 months after inclusion [15].

**Primary outcome.** Insomnia severity was measured using the insomnia severity index (ISI; [22]). The ISI consists of seven items that assess the nature, severity and the impact of insomnia symptoms (see S1 Table in S1 Appendix). Items are scored on a 5-point Likert scale, with higher scores indicating more severe insomnia symptoms. The ISI has good internal consistency (Cronbach's $\alpha = 0.78$) and is sensitive to changes in perceived sleep difficulties over time [23].

**Predictors.** As possible predictors of response shift effects we included: the trial type (insomnia only trial, versus insomnia with comorbid depression trial), age, anxiety (as measured with the Anxiety subscale of the Hospital Anxiety and Depression Scale (HADS-A; seven items on a four-point Likert scale, scores ranging 0–21, higher scores indicating higher levels of anxiety; [24]), and sleep-efficiency (percentage of sleep of the total time in bed; calculated from the online patients' sleep-diary records).

## Personal meaning for cancer survivors dataset

**Patients.** A total of 170 cancer survivors with a need for psychological support were randomly assigned to one of the three study arms: meaning-centred group psychotherapy for cancer survivors to improve personal meaning (MCGP-CS; n = 57), supportive group psychotherapy (SGP; n = 56), or control group (n = 57; care-as-usual control group without group intervention). As our aim was to investigate the process of change induced by the primary psychological treatment as compared to the control group, we only included data from patients in the MCGP-CS and control groups. Assessments were performed before randomization (baseline) and 9 weeks after the intervention or waiting period (post-assessment); questionnaires were filled out online or via pen and paper based on patients preference. In the treatment group, a total of 45 participants (79%) fulfilled the criteria of 'completers' (followed at least 4 out of 8 sessions; as completion of at least 4 sessions was considered to be sufficient to transfer the essential elements of the MCGP-CS treatment), and all participants completed post-assessment. Participants in the control group did not receive alternative treatments, but 13 participants did not complete post-assessment, thus a total of 44 participants had complete data. For more information regarding the methods of this trial see [16, 18].

**Intervention.** The main purpose of MCGP-CS is to sustain or enhance a sense of meaning or purpose in a patient's life, in order to cope better with the consequences of cancer. Theoretically, enhanced meaning is considered to be the catalyst of positive psychological outcomes. MCGP-CS is a manualized eight-week intervention that makes use of didactics such as a

personal workbook called 'Life lessons portfolio' containing homework-exercises, group discussions and experiential exercises that focus around themes related to meaning and cancer survivorship. The sessions lasted two hours each and were held weekly. The participants used the Life lessons portfolio to complete homework assignments every week. MCGP-CS was led by a psychotherapist with considerable experience in treating patients with cancer. Each session addressed a theme related to the concepts and sources of meaning, for example 'meaning before and after cancer', or 'attitudinal sources of meaning: encountering life's limitations'.

**Primary outcome.**   Personal meaning was assessed with the personal meaning profile-Dutch version (PMP-DV) [25]. The PMP-DV consists of 39 items scored on a 7-point Likert scale, that yield the following five subscales: relation with God/higher order (RG; 8 items), dedication to life (DL; 11 items), fairness of life (FL; 8 items), goal orientation (GO: 6 items), and relations with others (RO; 6 items) (see S4 Table in S1 Appendix for an overview of the items per subscale). The subscale scores are computed as mean item scores, with higher scores indicating more personal meaning.

**Predictors.**   Age and religiousness (yes/no) were included as possible predictors for the detected response shift.

### CBT for depressive symptoms in patients with diabetes dataset

**Patients.**   A total of 255 adult persons with diabetes with elevated depressive symptoms were randomly assigned to an internet-based CBT treatment (n = 130) or a control group (n = 125; waitlist control group). Assessments were performed online, before randomization (baseline) and 8 weeks after the intervention or waiting period (post-assessment). A total of 62 participants (48%) in the treatment group fulfilled the criteria for 'completers' (completed at least 5 out of 8 sessions; because all core concepts were explained in sessions 1 to 5), of whom 54 participants completed post-assessment. From the control group a total of 117 participants did not follow other psychological treatment, and 92 participants also completed post-assessment. For more information regarding the methods of this trial see [17].

**Intervention.**   Participants individually went through eight consecutive online lessons that provided written and spoken information and videos of diabetes patients with depression explaining how they learned from the course. Coaches (certified health psychologists) provided feedback on homework assignments. Feedback was to a large degree standardized and consisted of a concise, constructive reply on the applied CBT techniques, and meant to help patients understand and apply the CBT skills in daily practice. Participants allocated to the waiting list control group were offered the Web-based intervention if they still had elevated depressive symptoms 12 weeks after randomization.

**Primary outcome.**   Symptoms of depression were assessed with the centre for epidemiological studies depression scale (CES-D), a widely used 20-item self-report instrument. Respondents are asked to indicate the frequency with which they experienced depressive symptoms in the preceding week. The total score ranges from 0 to 60, with higher scores indicating more severe depression and scores of >16 representing a clinically significant level of depressive symptoms. Previous research [26, 27] has shown that the items can be grouped into four distinct aspects of depression, namely: depressed affect (DA; 7 items), well-being (WB; 4 items), somatic symptoms (SS; 7 items), and interpersonal problems (IP; 2 items) (see S5 Table in S1 Appendix for an overview of the items per subscale). For analyses in the current paper, we use these four subscale scores (calculated as mean item scores), where higher scores indicate higher levels of depressive symptoms.

**Predictors.**   As possible predictor of detected response shift effects we included diabetes distress at baseline as measured with the Dutch version of the problem areas in diabetes

(PAID) scale, a widely used 20-item self-report questionnaire [28]. Items pertain to negative emotions related to living with diabetes, rated on a 5-point Likert scale, ranging from 0 ("not a problem") to 4 ("a serious problem"). Sum scores are converted to a 0–100 scale, with higher scores indicating higher distress.

## Statistical analyses

Structural Equation Modelling (SEM) was used to investigate response shift in the primary outcome variable for both treatment- and control groups in all three datasets. The SEM approach for the investigation of response shift [29] uses a factor modelling framework to model the relations between observed variables (i.e. questionnaire scores) and the construct that the observed variables aim to measure (i.e. insomnia severity, personal meaning and depression respectively). When the factor model is applied to data from both baseline and follow-up measures, change in specific model parameters can be used to operationalize response shift effects that can be distinguished from change in the underlying construct of interest. Moreover, other variables can be included in the model to explain possible response shifts. The SEM procedure to investigate the occurrence of response shift and its possible impact on treatment effects included the following five steps: (1) establishing a measurement model, (2) overall test of response shift, (3) detection of specific response shift, (4) assessment of change in the construct of interest while taking into account possible response shift, and (5) inclusion of clinical- and background variables to predict the occurrence of response shift. It was used to test the three objectives: (1) the occurrence of response shift (steps 1–3), (2) the impact of response shift on the treatment effects (step 4), and (3) the predictive role of background variables. All statistical analyses were performed for each of the three datasets separately, with multigroup models to investigate and compare change and possible response shift in both the treatment and control groups. Data- and syntax files that were used for the reported analyses are available from the first authors' Open Science Framework page at https://osf.io/fv8c5/.

**Objective 1: The occurrence of response shift.** In the first step of the SEM procedure, one has to establish an interpretable and well-fitting "measurement model" in which the relationships between the observed scores and the underlying latent variables are specified. For all three primary outcome variables we used a one-factor model, where the underlying target construct (i.e. insomnia severity, personal meaning, and depression) is measured by the observed item- or subscale-scores from the associated questionnaire, both at baseline and follow-up assessment, for both treatment and control groups separately (see Fig 1). The model-fit of the measurement model was evaluated using the chi-square test of exact fit, and the root mean square error of approximation (RMSEA) as a measure of approximate fit. A non-significant chi-square test indicates exact model fit. An RMSEA value above 0.10 indicates 'poor' fit, below 0.08 indicates 'reasonable' fit and one below 0.05 'close' fit [30]. To inspect possible indications of model misfit we used modification indices and standardized residuals [31]. The power to detect a meaningful level of model misspecification of the measurement model was calculated based on the RMSEA [32]. The power to reject a measurement model with poor fit (i.e. RMSEA > 0.10) in favour of close fit (i.e. RMSEA < 0.05) for the three selected datasets are 1.0, 0.80 and 0.82 respectively.

The second step of the SEM procedure entails the specification of a model where all measurement parameters associated with response shift (i.e., factor loadings and intercepts) are constrained to be equal across baseline and follow-up assessments. The model fit of the resulting "no response shift model" can be compared to the model fit of the measurement model, using the difference in chi-square test statistics. Significant deterioration in model fit indicates the mere presence of overall response shift (any type of response shift). We also continued our

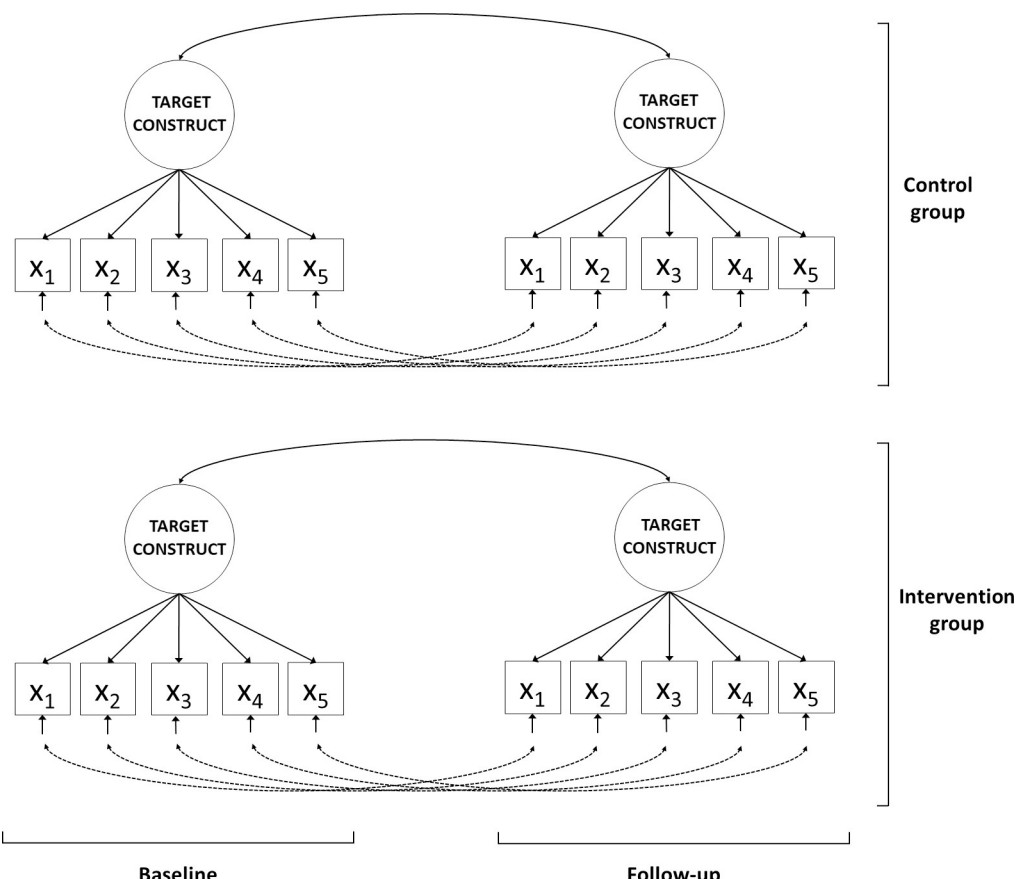

**Fig 1. An example of the measurement model used for response shift investigation.** *Notes*: The squares represent observed variables, i.e., questionnaire scores (X), measured at both baseline and follow-up assessment. The solid single-headed arrows at the bottom represent the residual factors of each observed variable. The dotted double-headed arrow represents the relations between the residual factors, where the residual factors of the same observed variable are allowed to correlate over time. The circles represent the target construct that the observed variables aim to measure (e.g., insomnia severity, depression, or personal meaning; both at baseline and follow-up assessment). Each arrow from a circle to an observed variable represents a factor loading. The double-headed arrows between the circles represent the correlations between the target construct over time.

investigation of response shift when the omnibus test for response shift was not significant, in order not to miss small but meaningful effects. To illustrate, the power to detect a difference in intercept of 0.5 (corresponding to a medium Cohen's *d* effect-size) and a difference in factor loading of 0.3 (corresponding to a medium Cohen's *r* effect-size) is only 0.55, 0.37, and 0.46 for the omnibus test in the three datasets respectively, whereas the power to detect such effects increases to 0.94, 0.74 and 0.77 when these effects would be tested one-by-one (see Appendix for details on power calculations). We note, however, that power calculations in the context of SEM are complicated and return to this issue in the Discussion.

The third step of the SEM procedure is used to investigate which variable is affected by which type of response shift. A change in the pattern of factor loadings (i.e., when a factor loading becomes zero) is taken as evidence of reconceptualization, as this indicates that the associated observed variable is no longer relevant to the measure of the target construct. A change in the value of factor loadings is taken as evidence of reprioritization, where a lower or higher value of the factor loading at follow-up assessment indicates that the associated observed variable has become less or more important to the measure of the target construct

respectively. Finally, a change in the intercepts is taken as evidence of recalibration, where lower or higher values of the intercept at follow-up indicate that the same level of the target construct is associated with lower or higher values of the associated observed variable respectively. We used an iterative procedure to indicate specific response shift, where we tested all possible model modifications (i.e., all types of response shift) in each iteration using the difference in chi-square test statistics. A significant chi-square difference test indicates that inclusion of the specific response shift effect significantly improves model fit. The response shift effect associated with the model modification that showed the largest (significant) improvement in model fit was included in the model. When different model modifications showed equivalent improvement in model fit, we used substantive considerations to choose between effects, in order to arrive at a sensible model. When response shift was evidenced, it was investigated whether response shift was present to the same extent in treatment versus control groups. The final model, that includes all possible indications of response shift, is referred to as the "response shift model".

**Objective 2: Impact of response shift on interpretation of treatment effects.** The impact of response shift on the interpretation of change can be evaluated at two levels. First, the impact of response shift on the estimated change in the variable for which response shift was detected can be calculated using a decomposition of change [33]. The decomposition entails that observed change is decomposed into change due to change in the underlying target construct and change due to response shift. Second, in the final model from step 3, the change in means of the underlying latent factors is indicative of change in the target construct. Comparison of the estimated change in the target construct from the final model with the estimated change in the no response shift model gives an indication of the impact of response shift on the overall treatment effect. For both types of impact of response shift we calculated SEM-based effect-size indices of change, where values of 0.3, 0.5 and 0.8 are indicative of 'small', 'moderate', and 'large' effects respectively [31].

**Objective 3: The predictive role of clinical and background variables.** In the fifth step of the SEM procedure, we extended the model to include possible predictors for detected response shift. The prediction model includes direct effects of the predictive variable on the item(s) or scale(s) that showed response shift. When an effect of a predictive variable is significant, this indicates that the predictor is associated with the detected response shift. Standardized effects of predictive variables can be interpreted as correlation coefficients, where values of 0.1, 0.3 and 0.5 are indicative of 'small', 'moderate', and 'large' effects respectively [31].

All statistical analyses were performed using the freely available software package Lavaan [34] in R [35]. Syntaxes of all statistical analyses reported in this paper are available as online supplementary material.

## Results

Tables 1–3 contain observed sample means and standard deviations of the primary outcomes and predictors at baseline and follow-up that were used for response shift analyses for each of the three datasets respectively.

### CBT for insomnia dataset

**Objective 1.** The multi-group longitudinal measurement model (see S1 Fig in S1 Appendix) with an added residual covariance between item 5 and item 6 showed the best fit (see Table 4). As expected, the no response shift model showed a significant deterioration in model fit as compared to the measurement model ($\Delta\chi^2$ (24) = 45.30, $p$ = .005), indicating the overall presence of response shift. The equality restriction on the intercept of item 2 ("Difficulty

**Table 1. Means and standard deviations of the primary outcome and predictors used for response shift analyses of the CBT for insomnia dataset.**

| CBT for insomnia dataset | | | | |
|---|---|---|---|---|
| | Control group (n = 77) | | CBT group (n = 86) | |
| | Baseline | Follow-up | Baseline | Follow-up |
| | M(SD) | M(SD) | M(SD) | M(SD) |
| *Primary outcome*: Insomnia severity[1] | | | | |
| Difficulty falling asleep | 2.16 (1.59) | 1.77 (1.45) | 2.12 (1.42) | 0.95 (1.05) |
| Difficulty staying asleep | 3.12 (1.01) | 3.00 (1.04) | 3.17 (1.03) | 1.38 (0.97) |
| Problems waking up too early | 2.62 (1.32) | 2.51 (1.28) | 2.38 (1.50) | 1.19 (1.15) |
| Satisfaction with sleep pattern | 3.63 (0.49) | 3.17 (0.80) | 3.44 (0.68) | 1.95 (1.00) |
| Interference daily functioning | 2.91 (0.66) | 2.51 (0.95) | 2.88 (0.74) | 1.55 (0.97) |
| Noticeable impact on QoL by others | 1.90 (0.85) | 1.67 (0.96) | 2.03 (0.87) | 1.12 (1.03) |
| Worries about sleep problems | 2.85 (0.77) | 2.66 (0.97) | 2.77 (0.81) | 1.34 (0.93) |
| *Predictors* | | | | |
| Age | 49.30 (14.87) | | 48.65 (13.90) | |
| Anxiety[2] | 7.43 (2.83) | | 7.87 (3.13) | |
| Sleep-efficiency | 69.22 (12.78) | | 70.24 (12.25) | |

*Notes*

[1] measured with the insomnia severity index (ISI)

[2] measured with the anxiety subscale of the hospital anxiety and depression scale (HADS-A).

staying asleep") was found not to be tenable ($\Delta\chi^2$ (2) = 17.26, $p < .001$), indicating the presence of recalibration response shift. As expected, this recalibration response shift was only significant in the CBT group ($\Delta\chi^2$ (1) = 14.98, $p < .001$) and not in the control group ($\Delta\chi^2$ (1) = 2.28, $p = .131$). Inspection of parameter estimates showed that the intercept "Difficulty staying asleep" was lower at follow-up as compared to baseline (see S1 Fig in S1 Appendix). In terms of response shift, this indicates that patients in the treatment group may have changed their interpretation of the response scale on this item such that it became easier to score lower on this item after treatment as compared to before treatment, given the same insomnia severity.

**Table 2. Means and standard deviations of the primary outcome and predictors used for response shift analyses of the personal meaning for cancer survivors dataset.**

| Personal meaning for cancer survivors dataset | | | | |
|---|---|---|---|---|
| | Control group (n = 44) | | MCGP-CS group (n = 45) | |
| | Baseline | Follow-up | Baseline | Follow-up |
| | M(SD) | M(SD) | M(SD) | M(SD) |
| *Primary outcome*: Personal meaning[1] | | | | |
| Relation with God/higher order | 2.84 (1.30) | 2.69 (1.36) | 2.87 (1.50) | 3.11 (1.60) |
| Dedication to life | 5.12 (0.91) | 4.89 (1.02) | 5.12 (0.95) | 5.27 (0.89) |
| Fairness of life | 4.34 (0.83) | 4.28 (1.02) | 4.43 (0.85) | 4.70 (0.79) |
| Goal-orientedness | 5.33 (1.02) | 4.79 (1.40) | 5.25 (1.09) | 5.58 (1.01) |
| Relation with other people | 5.77 (1.05) | 5.65 (1.24) | 5.38 (1.47) | 5.43 (1.32) |
| *Predictors* | | | | |
| Age | 56.71 (10.32) | | 58.62 (10.31) | |
| Religious (%) | 24 (55%) | | 18 (40%) | |

*Notes*

[1] measured with the personal meaning profile (PMP).

**Table 3. Means and standard deviations of the primary outcome and predictor used for response shift analyses of the CBT for depressive symptoms in patients with diabetes dataset.**

| CBT for depressive symptoms in patients with diabetes dataset | Control group (n = 92) | | CBT Group (n = 54) | |
|---|---|---|---|---|
| | Baseline | Follow-up | Baseline | Follow-up |
| | M(SD) | M(SD) | M(SD) | M(SD) |
| *Primary outcome*: *Depressive symptoms*[1] | | | | |
| Well-being | 1.89 (0.55) | 1.65 (0.58) | 1.94 (0.56) | 1.28 (0.70) |
| Depressive affect | 1.19 (0.47) | 0.87 (0.48) | 1.20 (0.53) | 0.62 (0.54) |
| Somatic symptoms | 1.52 (0.47) | 1.25 (0.52) | 1.58 (0.50) | 0.96 (0.54) |
| Interpersonal problems | 0.71 (0.65) | 0.57 (0.56) | 0.88 (0.62) | 0.50 (0.59) |
| *Predictor* | | | | |
| Diabetes distress[2] | 30.7 (15.1) | | 34.5 (15.0) | |

*Notes*

[1] measured with the centre for epidemiological studies depression scale (CES-D)

[2] measured with the problem areas in diabetes (PAID) scale.

**Objective 2.** The detected response shift had a moderate impact on the change in item 2, where the observed reduction in this item score was larger than what would have been expected if the item response scale was equivalent as compared to baseline (response shift effect, Cohen's $d$ = -0.75; see also S1 Table in S1 Appendix). The decrease in insomnia severity in the CBT group was somewhat smaller when taking into account response shift ($d$ = -1.75) as compared to when response shift was not taken into account ($d$ = -1.92). Hence, other than expected, taking into account response shift only slightly reduced the overall treatment effect.

**Objective 3.** Anxiety and sleep-efficiency showed small but significant associations with the detected recalibration response shift ($r$ = -.18, $p$ = .047; $r$ = -.22, $p$ = .009), indicating that more anxiety and better sleep-efficiency are associated with more recalibration response shift in item 2. That is, patients who score higher on anxiety and sleep-efficiency at baseline tend to more easily indicate less problems with difficulty staying asleep at follow-up. There were no significant effects of age or the type of trial (i.e., the insomnia trial versus the insomnia and depression trial).

**Table 4. Overall goodness of fit of the models in steps 1–3 of the SEM-approach for investigation of response shift (objective 1) for each of the three datasets.**

| | Chi-square | Df | p-value | RMSEA [90% CI] | CFI |
|---|---|---|---|---|---|
| **CBT for Insomnia Dataset** | | | | | |
| Step 1: Measurement model | 227.73 | 138 | < .001 | 0.089 [0.068–0.109] | 0.895 |
| + residual covariance item 5–6 | 189.30 | 130 | .001 | 0.075 [0.050–0.096] | 0.930 |
| Step 2: No response shift model | 234.60 | 154 | < .001 | 0.080 [0.059–0.100] | 0.905 |
| Step 3: Response shift model | 219.62 | 153 | < .001 | 0.073 [0.050–0.094] | 0.922 |
| **Personal Meaning for Cancer Survivors Dataset** | | | | | |
| Step 1: Measurement model | 76.67 | 58 | .052 | 0.085 [0.000–0.133] | 0.975 |
| Step 2: No response shift model | 111.72 | 74 | .040 | 0.107 [0.063–0.146] | 0.949 |
| Step 3: Response shift model | 98.45 | 72 | .009 | 0.091 [0.037–0.133] | 0.956 |
| **CBT for Depressive Symptoms in Patients with Diabetes Dataset** | | | | | |
| Step 1: Measurement model | 24.83 | 30 | .733 | 0.000 [0.000–0.067] | 1.000 |
| Step 2: No response shift model | 39.02 | 42 | .603 | 0.000 [0.000–0.071] | 1.000 |
| Step 3: Response shift model | 30.38 | 40 | .867 | 0.000 [0.000–0.044] | 1.000 |

## Personal meaning for cancer survivors dataset

**Objective 1.** The multi-group longitudinal measurement model (see S2 Fig in S1 Appendix) showed good fit (see Table 4). As expected, the no response shift model showed a significant deterioration in model fit as compared to the measurement model ($\Delta\chi^2$ (16) = 35.05, $p$ = .004), indicating the overall presence of response shift. The equality restriction on the factor loading of the subscale RG ("relations to God/higher order") was found not to be tenable ($\Delta\chi^2$ (2) = 13.17, $p$ = .001), indicating the presence of reprioritization response shift. Unexpectedly, this reprioritization response shift was significant in both the MCGP-CS group ($\Delta\chi^2$ (1) = 5.73, $p$ = .017) as well as in the control group ($\Delta\chi^2$ (1) = 7.54, $p$ = .006). However, inspection of parameter estimates showed that the effect was in opposite direction. The factor loading of the subscale RG was higher after treatment in the MCGP-CS group as compared to baseline, whereas the value of the same factor loading in the control group decreased over time (see S2 Fig in S1 Appendix). In terms of response shift, this indicates that for patients in the treatment group the subscale RG became more important to the measurement of personal meaning after treatment, whereas the same subscale became less important to patients in the control group.

**Objective 2.** The detected response scale had a very small impact on the estimated change in the subscale RG. The increase in this subscale was larger than what would be expected if the item was equally important as compared to baseline (response shift effect, Cohen's $d$ = 0.05; see also S2 Table in S1 Appendix), whereas the decrease in this subscale was smaller than expected due to a decrease in importance of the subscale in the control group (Cohen's $d$ = 0.06). The increase in personal meaning in the MCGP-CS group was of comparable size with ($d$ = 0.52) and without taking response shift into account ($d$ = 0.53). Also, the decrease in personal meaning in the control group was only marginally different with ($d$ = -0.50) and without ($d$ = -0.52) taking into account response shift. Thus, other than expected, the detected response shift did not impact the treatment effect.

**Objective 3.** In both the treatment- and control groups there was a small but significant association of religiousness with the detected reprioritization response shift in the subscale RG (MCGP-CS group: $r$ = .17, $p$ = .007; control group: $r$ = .19, $p$ = .026;). These results indicate that being religious before the start of treatment is associated with more reprioritization response shift in the treatment group (i.e., relation with god/higher order becomes more important, and this is especially true for religious people), and with less reprioritization response shift in the control group (i.e., relation with god/higher order becomes less important, but this is especially true for *non*-religious people). There were no significant effects of age.

## CBT for depressive symptoms in patients with diabetes dataset

**Objective 1.** The multi-group longitudinal measurement model (see S3 Fig in S1 Appendix) showed good fit (see Table 4). The no response shift model did not show a significant deterioration in model fit as compared to the measurement model ($\Delta\chi^2$ (12) = 13.71, $p$ = .319). Other than expected, this indicates there is no overall presence of response shift. To prevent missing small but meaningful effects, we continued the investigation for possible specific indications of response shift. The equality restriction on the factor loading of the subscale DA ("Depressive Affect") was found not to be tenable ($\Delta\chi^2$ (2) = 8.63, $p$ = .013), indicating the presence of reprioritization response shift. Unexpectedly, this reprioritization response shift was significant in both the CBT group ($\Delta\chi^2$ (1) = 4.45, $p$ = .035), and the control group ($\Delta\chi^2$ (1) = 4.18, $p$ = .041). Inspection of parameter estimates showed that the factor loading of DA became smaller at follow-up, indicating that the subscale depressive affect became less important to the measurement of depression in both groups (see S3 Fig in S1 Appendix).

**Objective 2.** The detected response shift had a moderate to large impact on the change in the subscale DA, where the estimated change was smaller than what would be expected if the subscale was equally important as compared to baseline (response shift effect, Cohen's $d = 0.46$ (control group) and $d = 0.96$ (CBT group); see also S3 Table in S1 Appendix). The decrease in depression in the CBT group was only slightly larger with ($d = -1.58$) as compared to without taking response shift into account ($d = -1.51$). Also, the decrease in depression in the control group was slightly larger with ($d = -0.80$) as compared to without ($d = -0.77$) taking into account response shift. Thus, other than expected, the detected response shift did only marginally impact the treatment effect.

**Objective 3.** We found no significant association between diabetes distress and the detected reprioritization response shift of the subscale DA, indicating that diabetes distress at baseline is not related to a change in the importance of the DA subscale.

## Discussion

In the current study, we investigated whether response shift induced by psychological interventions to improve health impacts the interpretation of its results. In all secondary analyses of the three datasets, we found indications of response shift, however, these response shifts only marginally impacted trial results.

### Detected response shift and impact on trial results

In the CBT for insomnia dataset, we found one indication of recalibration response shift in 'difficulty staying asleep' induced by treatment (i.e., present in the treatment condition only), that moderately impacted change in the item for which response shift was detected. Taking into account response shift marginally reduced the treatment effect. Associations with anxiety and sleep-efficiency showed that the detected recalibration response shift occurred most strongly in patients who scored high on anxiety or sleep-efficiency before the start of treatment.

In the personal meaning for cancer survivors dataset, we found one indication of reprioritization response shift for 'relation with god/higher order' in both treatment and control groups, but in opposite directions. Occurrence of the detected reprioritization response shift was found to be associated with religiousness. However, the impact of the detected response shift on change was negligible both at the level of the subscale and at the level of the overall treatment effect.

In the CBT for depressive symptoms in patients with diabetes dataset we found one indication of reprioritization response shift of 'depressive affect' in both treatment and control group, which was not found to be associated with selected predictors. Although the detected response shift had moderate (control group) to large (treatment group) impacts on change in 'depressive affect', it did not impact overall trial results. The occurrences of response shift thus provide a more detailed description of the changes that occur, i.e., how treatment affects the different aspects of the primary self-reported outcomes differently, but does not influence the overall trial conclusions. In the following, we address these occurrences of response shift and how they are clinically meaningful.

### Interpretation and explanation of response shift

In the CBT for insomnia dataset, we found that patients in the treatment group more easily reported to have less "difficulty staying asleep" after treatment as compared to before treatment. Patients may have changed their interpretation of the response scale due to a re-evaluation of the severity of this particular symptom; for example, patients may learn to consider

waking up at the end of a sleep-cycle to be part of a normal and healthy sleep routine. Due to this re-evaluation, the observed reduction in scores on the difficulty staying asleep item amplifies. Interestingly, the same item was found to be one of the two aspects of insomnia impacted most consistently by treatment in a network analysis applied to part of the same data [36]. These combined results illustrate how response shift analyses may help to understand treatment effectiveness, in that it shows that treatment induces specific (amplifying) effects on specific aspects of insomnia.

We also found that individuals who experienced more severe anxiety at the start of the treatment, and better sleep-efficiency, are especially prone to response shift on the 'difficulty falling asleep' item. This may indicate that individuals who tend to spend their time in bed more often sleeping, and who show anxious thoughts, also tend to more easily shift their interpretation of how difficult it is to fall asleep. For example, it may be that anxious patients are especially prone to benefit from education about normal sleep patterns (e.g., that waking up after a sleep-cycle is part of a normal sleep routine) that elicits the detected recalibration response shift.

Although there is some previous research on the occurrence of response shift in measures of fatigue (cf. [37–39]), those studies mostly pertain to fatigue in cancer patients and response shift induced by radio- or chemotherapy; and thus their results are difficult to compare to the results of our study, and not informative with regards to the interpretation of response shift due to psychological intervention. With regards to the latter, more research is needed to investigate the role of the described cognitive re-evaluation of the severity of difficulty staying asleep in explaining the detected response shift. For example, comparison of self-report evaluations about staying asleep and physical measurements (e.g., polysomnography or actigraphy) may help to interpret the relation between response shift in self-report and actual sleep routine. Also, investigation of response shift for insomnia treatments with specific cognitive components as compared to specific behavioural components could further shed light on whether response shifts can be considered as an aspect of treatment efficacy. Results from our study could be used to derive a-priori hypotheses about whether response shift is expected (e.g., only in treatments that include specific cognitive aspects), and how it may impact trial results (e.g., amplifying treatment effectiveness).

In the personal meaning for cancer survivors dataset, we found that relation with god or a higher order became more important to the measurement of personal meaning for patients in the treatment group, particularly for those who indicated to be religious, whereas it became less important to the measurement of personal meaning in the control group, particularly for those who indicated to be *non*-religious. Although we expected to only find response shift in the treatment group, this result shows a differential effect of response shift between treatment and control group. However, the detected response shifts had negligible impact on the treatment effect. It could be that the detected effects in this study were small because the included respondents had relatively mild symptoms as they were not selected based on the severity of their symptoms but on their need for psychological support. Replication of results is needed to corroborate the current findings. Future research could investigate response shift in similar trials of meaning-centred psychotherapy that use personal meaning as a primary outcome, and that include patients who report higher levels of distress. Also, it may be interesting to further explore the relation between religiousness and personal meaning by including more detailed information about the type and/or practices of religion. It may be, for example, that certain aspects of the psychological intervention are especially beneficial for the improvement of personal meaning in specific religious groups.

In the CBT for depressive symptoms in patients with diabetes dataset, we found that "depressive affect" became less important to the measurement of depression. If this effect would have occurred only in the treatment group, it might have been an adaptive treatment-

induced effect, where patients learn to accept symptoms of depressive affect and as a result interpret them as less important. However, this response shift effect was also found, although to a lesser degree, in the control group. This suggests that the adaptive effect is likely–at least partly–due to a natural course of depression rather than (solely) a result of treatment (see for example [40]). The detected response shift was not found to be associated with diabetes distress, and only marginally increased overall treatment effectiveness. Occurrence of response shift in measures of depression has been evidenced in previous studies too [10–13, 41]. Similar to our findings, some studies indicated response shift in measures like depressive affect, where Carlier [13] detected reconceptualization of mood items, Smith et al. [12] found reprioritization of the item 'how often do you feel depressed', and Wu [11] found considerable recalibration response shift in items of the negative attitude factor. Interestingly, these three studies investigated response shift after psychological intervention. The results of these studies combined suggest that re-interpretation of aspects of depressive affect may be part of the effect of psychological intervention. However, these studies vary in the type of depression measure that was investigated, the type of response shift that was detected, the direction of response shift effects and resulting impact on treatment effectiveness. Therefore, these results need to be corroborated. For example, with respect to the results of the current study, it would be interesting to investigate how patients interpret measures of depressive affect both during the natural course of time and after specific aspects of (cognitive) treatment.

## Strengths and limitations

Strengths of the current proof-of-principle study into psychological intervention-induced response shifts is the combination of different trials, employing varying patient populations, intervention types, and primary outcome measures. This heterogeneity highlighted differences in the response shift findings across trials. This finding may let us hypothesize that for example some constructs or interventions are more sensitive to response shift than others. Finally, by including predictors of response shift the current study went beyond merely showing the occurrence of response shift, but also aimed to provide a clinically meaningful interpretation of the detected effects.

A limitation of the current study is that the sample size for each individual dataset was not large. Limited sample sizes raise the question of whether there was enough power to detect meaningful effects. Power calculations for SEM-analyses are complicated, because the effect-size that needs to be specified depends on many parameters in the model. Although there exists general rules of thumb that recommend a minimum absolute sample size (e.g., N = 100 or 200 or a minimum number of observations per variable), these rules generally have little empirical support (cf. [42, 43]). One reason is that complexity of the model (e.g. number of underlying factors, and number of indicators per factor), and strength of associations (e.g. factor loadings and factor correlations) are also important to consider (cf. [44]). For overall model fit evaluation, we therefore relied on RMSEA-based power calculation, which has the advantage that the effect-size is based on the specification of associated RMSEA-values instead of all parameters in the model. For the detection of response shift effects, we used a chi-square based power calculation, thus relying on the specification of all parameters in the model. As the effect-size (and thus power) depends on the chosen values in these model specifications, other values may lead to other conclusions regarding the achieved power. We included these calculations for illustrative purposes only. In addition to statistical significance, effect-sizes are important for an evaluation of the relevance of detected effects.

Another limitation of the current study is that we used different levels of analyses for the different measurement instruments in the respective trials. That is, for the CBT for insomnia

dataset we used item-level analyses to investigate response shift, whereas in the personal mean-ing for cancer survivors dataset and the CBT for depressive symptoms in patients with diabetes dataset, we used subscale-level analyses. Such differences in the modelling strategy may have impacted results, as possible response shift at the item-level may be obfuscated at the subscale level (e.g. response shifts at the item-level may cancel each other out). We have chosen this modelling strategy because we thought it important to use the level of analyses that is consis-tent with the level of analyses and interpretation in the published RCTs and/or as is commonly used for the respective measurement instruments That is, the insomnia questionnaire is usu-ally interpreted as an overall score in relation to individual item scores (consistent with item-level analyses), whereas the personal meaning and depression questionnaires are more com-monly interpreted as an overall score in relation to the subscale scores (consistent with sub-scale-level analyses).

Based on the considerations above, results from the individual datasets thus need to be interpreted with caution. Further research is needed to corroborate results from the current study and to better explain the mechanisms of response shift, possibly by including other pre-dictors, or by using other–qualitative–research methods.

## Conclusions

This study indicated that response shift effects can occur as a result of psychological interven-tions to improve health in different patient groups and a variety of self-report measures. More importantly, this study showed that occurrences of response shift did not influence the overall interpretation of treatment effectiveness, thus supporting the trial results. Nevertheless, inves-tigation of response shift can provide insight into differential treatment effectiveness for spe-cific symptoms and/or domains that can be clinically meaningful. The results of this study can be used to generate hypotheses about occurrences of response shift and their clinical impact, which need to be corroborated in future explorations of response shift.

## Supporting information

**S1 Appendix. Detailed results of the response shift investigation per dataset.**
(DOCX)

**S2 Appendix. Chi-square based power calculations for the tests on response shift.**
(R)

## Author Contributions

**Conceptualization:** M. G. E. Verdam, H. Knoop, F. J. Oort, H. Riper, A. van Straten, I. M. Verdonck-de Leeuw, M. A. G. Sprangers.

**Data curation:** M. G. E. Verdam, W. van Ballegooijen, M. de Wit.

**Formal analysis:** M. G. E. Verdam.

**Funding acquisition:** M. G. E. Verdam, H. Knoop, F. J. Oort, H. Riper, A. van Straten, I. M. Verdonck-de Leeuw, M. A. G. Sprangers.

**Investigation:** C. J. M. Holtmaat, J. Lancee, T. van der Zweerde.

**Methodology:** F. J. Oort.

**Project administration:** M. G. E. Verdam, M. A. G. Sprangers.

**Resources:** H. Riper, A. van Straten, I. M. Verdonck-de Leeuw.

**Supervision:** M. A. G. Sprangers.

**Visualization:** M. G. E. Verdam.

**Writing – original draft:** M. G. E. Verdam.

**Writing – review & editing:** W. van Ballegooijen, C. J. M. Holtmaat, H. Knoop, J. Lancee, F. J. Oort, H. Riper, A. van Straten, I. M. Verdonck-de Leeuw, M. de Wit, T. van der Zweerde, M. A. G. Sprangers.

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
