## [Decision Letter · Decision Letter 0]

12 Feb 2021

PONE-D-21-02214

Re-evaluating randomized clinical trials of psychological interventions:

Impact of response shift on the interpretation of trial results

PLOS ONE

Dear Dr. Verdam,

Thank you for submitting your manuscript to PLOS ONE. After careful consideration, we feel that it has merit but does not fully meet PLOS ONE’s publication criteria as it currently stands. Therefore, we invite you to submit a revised version of the manuscript that addresses the points raised during the review process.

We look forward to receiving your revised manuscript.

Kind regards,

Ali Montazeri

Academic Editor

PLOS ONE

Journal Requirements:

Reviewers' comments:

Reviewer's Responses to Questions

**Comments to the Author**

1. Is the manuscript technically sound, and do the data support the conclusions?

Reviewer #1: No

Reviewer #2: Yes

2. Has the statistical analysis been performed appropriately and rigorously? 

Reviewer #1: I Don't Know

Reviewer #2: Yes

3. Have the authors made all data underlying the findings in their manuscript fully available?

Reviewer #1: Yes

Reviewer #2: Yes

4. Is the manuscript presented in an intelligible fashion and written in standard English?

Reviewer #1: Yes

Reviewer #2: Yes

5. Review Comments to the Author

Reviewer #1: Re-evaluating randomized clinical trials of psychological interventions: Impact of response shift on the interpretation of trial results.

The study re-analyses data of published randomized controlled trials (RCTs) investigating the effectiveness of psychological interventions targeting different health aspects, to assess the occurrence of response shift, the impact of response shift on interpretation of treatment effectiveness, and the predictive role of clinical and background variables for detected response shift.

- My major question is about the process of including RCTs to evaluate them in this study. How this was done ? what was the inclusion and exclusion criteria for selecting the studies. Did the authors search systemically through the Data bases to find appropriate ounces? Please, explain the process clearly.

-The methodology of study was written without specific structure and without coherence understandable way.

-I found that the study included Internet Based RCTs, please mention Internet based in Title, aims, and as inclusion criteria.

- I suggest the authors to describe data collection approaches. were they based on Internet or not. Please explain in detailed.

Reviewer #2: The manuscript investigated a pivotal issue in the field, which needs further attention. The presence of response shift is studied in three different psychological interventions. The findings concerning both the results and the interpretation are discussed. The manuscript is well-written and well-organized. Various aspects of concerns that I wondered about have been addressed. The methodology is presented clearly. And the study findings are discussed sufficiently. The implications are explained straightforward. Future directions are provided.

However, the first issue that might be worth noting is that although the sample sizes of individual RCTs are small to an almost equal degree, (especially for the treatment groups, 86, 45, and 54), the specifications of the measurement models are to some extent different. Therefore, it is encouraged to include a description of appropriate sample size for each model rendering the study power to be satisfactory.

In this respect, Christopher Westland has provided a functional way to determine an appropriate sample size based on the model specification (https://doi.org/10.1016/j.elerap.2010.07.003). Moreover, Wolf et al (2013 PMC4334479) has urged that how the strength of factor loading, among other factors, can affect the sample size requires for the power to be satisfactory. The authors may want to appropriately discuss (or explain in the methods sections) whether the individual sample sizes and their corresponding model specifications might affect the study results since, for example, 1) the model specifications are different, i.e. number of observed variables, 2) the measurement models differed in terms of some sample-size/power-related factors, such as the strength of factor loading.

Secondly, as it is reflected in the results of model fits, the differences in the modeling might be another source of concern that needs to be addressed. That is, the model of CBT for insomnia included ordinal indicators while in the other two models the aggregated scores are used. It could affect the presence of any evidence concerning the first study aim (i.e. the presence of response shift) since aggregating the ordinal scoring blur the measurement errors, hence the underlying assumptions of SEM. In addition, it is not clear that whether the applied methodology representing the “shift” differs in the different modeling strategies (i.e. ordinal versus aggregated indicators). Importantly, the CBT for insomnia model has used a unidimensional measurement model, while the other two models are bifactor models in nature. That is, the measured indicators in which the response shift is assumed to occur indicate the sub-scales, which altogether compose a higher-order construct. Thus, the authors are encouraged to base the measurement models on ordinal indicators and repeat the analysis using the subscales separately to see any noticeable findings and to maintain the applied methodology consistent for all models. Otherwise, more description is needed to address how the applied methodology could represent the response shift in interval-continuous indicators that were not directly asked from the participants (For example, Relation with God/higher order, and Depressed affect).

Thirdly, the applied scales were primarily used Likert-type scaling. The authors are encouraged to discuss how the scaling may affect the results, and address the other types of scaling such as semantic differential scaling.

Best of Luck,

6. PLOS authors have the option to publish the peer review history of their article (what does this mean?). If published, this will include your full peer review and any attached files.

Reviewer #1: No

Reviewer #2: No

---

## [Author Response · Author response to Decision Letter 0]

12 Apr 2021

PLOS ONE Journal Requirements:

Response: We have found and changed an error in the file naming Figure 1 (it was named as supporting information but should be included in the main manuscript). We did not find further deviations from PLOS ONE’s style requirements and trust that was all that was referred to here. 

Your ethics statement should only appear in the Methods section of your manuscript. If your ethics statement is written in any section besides the Methods, please delete it from any other section.

Response: We have now integrated the Ethics statement into the Methods section

Review Comments to the Author

Reviewer #1: Re-evaluating randomized clinical trials of psychological interventions: Impact of response shift on the interpretation of trial results.

The study re-analyses data of published randomized controlled trials (RCTs) investigating the effectiveness of psychological interventions targeting different health aspects, to assess the occurrence of response shift, the impact of response shift on interpretation of treatment effectiveness, and the predictive role of clinical and background variables for detected response shift.

- My major question is about the process of including RCTs to evaluate them in this study. How this was done? what was the inclusion and exclusion criteria for selecting the studies. Did the authors search systemically through the Data bases to find appropriate ounces? Please, explain the process clearly.

Response: The studies that were selected for the analyses in the current paper are a convenience sample, which was previously stated on page 7, first line in the Introduction section. The studies were selected based on the following criteria: 1) the RCT results indicate effectiveness of the psychological intervention; and thus 2) response shift may have occurred; 3) total number of included patients > 100; and 4) availability of clinical and background information that can be included as explanatory variables. We now included this information in the Method section on page 7 (first paragraph).

-The methodology of study was written without specific structure and without coherence understandable way.

Response: The method section of the paper indeed deviates from that of a single sample study. Since we need to describe three datasets (based on data from four RCTs) that were used for the analyses, we adopted a similar structure for each dataset, including the description of “Patients”, “Intervention”, “Primary Outcome”, and “Predictors”. After the description of the datasets follows a paragraph on “Statistical Analyses” that describes the methodology used to investigate the three objectives of the study applied to all three datasets. In doing so, we believe that the chosen structure is clear and coherent. 

-I found that the study included Internet Based RCTs, please mention Internet based in Title, aims, and as inclusion criteria.

- I suggest the authors to describe data collection approaches. were they based on Internet or not. Please explain in detailed.

Response: It is our aim to investigate the occurrence of response shift after a psychological intervention. Although three of the four included studies are indeed internet-based studies, this was not a specific selection criterion. We now include the selection-criteria of the included studies in the Method section on page 7 (first paragraph) to avoid confusion. In the Method section we now specify for each study whether assessments were performed online or via pen and paper. Since the way the interventions are delivered – face-to face or via internet – is not relevant to our purpose, we rather not include the term internet in the title nor aims as it may confuse and distract.

Reviewer #2: The manuscript investigated a pivotal issue in the field, which needs further attention. The presence of response shift is studied in three different psychological interventions. The findings concerning both the results and the interpretation are discussed. The manuscript is well-written and well-organized. Various aspects of concerns that I wondered about have been addressed. The methodology is presented clearly. And the study findings are discussed sufficiently. The implications are explained straightforward. Future directions are provided.

Response: Thank you for your positive remarks regarding our manuscript and your suggestions for improvement.

However, the first issue that might be worth noting is that although the sample sizes of individual RCTs are small to an almost equal degree, (especially for the treatment groups, 86, 45, and 54), the specifications of the measurement models are to some extent different. Therefore, it is encouraged to include a description of appropriate sample size for each model rendering the study power to be satisfactory.

In this respect, Christopher Westland has provided a functional way to determine an appropriate sample size based on the model specification ((https://doi.org/10.1016/j.elerap.2010.07.003). Moreover, Wolf et al (2013 PMC4334479) has urged that how the strength of factor loading, among other factors, can affect the sample size requires for the power to be satisfactory. The authors may want to appropriately discuss (or explain in the methods sections) whether the individual sample sizes and their corresponding model specifications might affect the study results since, for example, 1) the model specifications are different, i.e. number of observed variables, 2) the measurement models differed in terms of some sample-size/power-related factors, such as the strength of factor loading.

Response: The reviewer raises an important issue with regard to the differences in sample-size and model specification and the effects on power. We agree with the reviewer that an evaluation of the power to detect a meaningful level of model misspecification (for the specification of the measurement model) and the power to detect response shift effects is important. However, power calculation in the context of SEM are complicated. In order to address this issue we now do include RMSEA-based power calculations for the specified measurement models (page 15, final paragraph) and chi-square based power calculations for the tests on response shift (page 16, first paragraph). We include these calculations in an appendix and discuss the issue of power (calculations) in more detail in the Discussion section (page 32). 

Secondly, as it is reflected in the results of model fits, the differences in the modeling might be another source of concern that needs to be addressed. That is, the model of CBT for insomnia included ordinal indicators while in the other two models the aggregated scores are used. It could affect the presence of any evidence concerning the first study aim (i.e. the presence of response shift) since aggregating the ordinal scoring blur the measurement errors, hence the underlying assumptions of SEM. In addition, it is not clear that whether the applied methodology representing the “shift” differs in the different modeling strategies (i.e. ordinal versus aggregated indicators). Importantly, the CBT for insomnia model has used a unidimensional measurement model, while the other two models are bifactor models in nature. That is, the measured indicators in which the response shift is assumed to occur indicate the sub-scales, which altogether compose a higher-order construct. Thus, the authors are encouraged to base the measurement models on ordinal indicators and repeat the analysis using the subscales separately to see any noticeable findings and to maintain the applied methodology consistent for all models. Otherwise, more description is needed to address how the applied methodology could represent the response shift in interval-continuous indicators that were not directly asked from the participants (For example, Relation with God/higher order, and Depressed affect).

Thirdly, the applied scales were primarily used Likert-type scaling. The authors are encouraged to discuss how the scaling may affect the results, and address the other types of scaling such as semantic differential scaling.

Response: We agree with the reviewer that the differences in the modelling of the different datasets may have impacted results. It is true that when investigating response shift at the subscale level, possible response shift at the item-level may be obfuscated. For example, response shifts at the item-level may cancel each other out at the subscale level. There are two reasons for using the modelling strategies as described. First, using item-level analyses for the personal meaning and depression datasets would have led to a substantive increase in the complexity of the model that – considering the limited sample size – would have caused problems with convergence. Second, and more importantly, we wanted the level of analyses to be consistent with the level of analyses as in the published RCTs and/or commonly used interpretation of the measurement scales. That is, the insomnia questionnaire is usually interpreted as an overall score in relation to individual item scores (consistent with the one-factor model of the item-scores), whereas the depression scale and personal meaning questionnaire are usually interpreted as an overall score in relation to the subscale scores (consistent with the one-factor model of the subscale-scores). To better address the limitations of the differences in technical specification of the models, and our reasons to do so, we have now added a paragraph in the Discussion section (pages 32-33).

---

## [Decision Letter · Decision Letter 1]

10 May 2021

Re-evaluating randomized clinical trials of psychological interventions:

Impact of response shift on the interpretation of trial results

PONE-D-21-02214R1

Dear Dr. Verdam,

We’re pleased to inform you that your manuscript has been judged scientifically suitable for publication and will be formally accepted for publication once it meets all outstanding technical requirements.

Kind regards,

Ali Montazeri

Academic Editor

PLOS ONE

Additional Editor Comments (optional):

Reviewers' comments:

Reviewer's Responses to Questions

**Comments to the Author**

1. If the authors have adequately addressed your comments raised in a previous round of review and you feel that this manuscript is now acceptable for publication, you may indicate that here to bypass the “Comments to the Author” section, enter your conflict of interest statement in the “Confidential to Editor” section, and submit your "Accept" recommendation.

Reviewer #2: All comments have been addressed

2. Is the manuscript technically sound, and do the data support the conclusions?

Reviewer #2: Yes

3. Has the statistical analysis been performed appropriately and rigorously? 

Reviewer #2: Yes

4. Have the authors made all data underlying the findings in their manuscript fully available?

Reviewer #2: Yes

5. Is the manuscript presented in an intelligible fashion and written in standard English?

Reviewer #2: Yes

6. Review Comments to the Author

Reviewer #2: I would like to thank you the authors for being responsive to my comments. I have found this study interesting and informative to the filed. The current version of the manuscript merits publication if the manuscript is found to adhere to the Journal’s style.

7. PLOS authors have the option to publish the peer review history of their article (what does this mean?). If published, this will include your full peer review and any attached files.

Reviewer #2: No

---

## [Editor Report · Acceptance letter]

17 May 2021

PONE-D-21-02214R1 

Re-evaluating randomized clinical trials of psychological interventions: Impact of response shift on the interpretation of trial results 

Dear Dr. Verdam:

I'm pleased to inform you that your manuscript has been deemed suitable for publication in PLOS ONE. Congratulations! Your manuscript is now with our production department. 

Kind regards, 

on behalf of

Professor Ali Montazeri 

Academic Editor

PLOS ONE